# Why Do Women Not Use the Bathroom? Women’s Attitudes and Beliefs on Using Public Restrooms

**DOI:** 10.3390/ijerph17062053

**Published:** 2020-03-20

**Authors:** Siobhan M. Hartigan, Kemberlee Bonnet, Leah Chisholm, Casey Kowalik, Roger R. Dmochowski, David Schlundt, W. Stuart Reynolds

**Affiliations:** 1Department of Urology, Vanderbilt University Medical Center, Nashville, TN 27232, USA; siobhan.hartigan@vumc.org (S.M.H.); leah.p.chisholm@vanderbilt.edu (L.C.); roger.dmochowski@vumc.org (R.R.D.); 2Department of Psychology, Vanderbilt University Medical Center, Nashville, TN 27232, USA; kemberlee.bonnet@vanderbilt.edu (K.B.); david.schlundt@vanderbilt.edu (D.S.); 3Department of Urology, University of Kansas Medical Center, Kansas City, KS 66160, USA; ckowalik@kumc.edu

**Keywords:** bladder health, lower urinary tract symptoms, restrooms, women’s health, qualitative research, bowel health

## Abstract

There are a variety of factors and influences, both internal and external, that may impact an individual’s public toileting experience and may ultimately have repercussions for bladder health. This study sought to identify predominant constructs underlying a women’s attitude towards using restrooms at work, at school, and in public in order to develop a conceptual model incorporating these themes. We performed a secondary analysis of a cross-sectional, survey-based study that included open-ended questions about limitations to restroom use using a mixed-methods approach. Qualitative data coding and analysis was performed on 12,583 quotes and, using an iterative inductive-deductive approach, was used to construct the conceptual framework. Our conceptual framework reveals a complicated interplay of personal contexts, situational influences, and behavioral strategies used by women to manage their bladder and bowel habits away from home. These findings can inform future research and public policy related to bladder health awareness related to toilet access in the workplace and in public.

## 1. Introduction

Most people become toilet trained during toddlerhood. By this stage in life, humans are able to recognize the feeling of a full bladder as the time to go to the bathroom and begin to learn social cues and mores related to toilet use. For most, urinating is an instinctual and routine act that occurs without much thought. However, the act of going to the bathroom or using a restroom to urinate is not nearly as simple as it may seem. In fact, there are multiple factors, both internal and external, that may impact an individual’s toileting experience and thus have ultimate repercussions for bladder health. For instance, people tend to avoid public restrooms because they may be inaccessible, may be unsanitary, or may raise safety concerns that discourage their use [1,2,3,4], and despite having the desire to urinate, many will delay using the restroom when they are away from home [2,3]. However, limiting restroom use and delaying voiding are considered unhealthy bladder habits that may ultimately lead to urinary dysfunction and poor bladder health [1,2,3,5].

Intuitively, most people understand that using public restrooms or restrooms at work may pose barriers or aversions. However, few studies have specifically studied external, environmental factors that contribute to limitations on the use of public restroom use or how these environmental factors may interact with internal perceptions of restroom need [2]. However, the social ecology and environment in which women exist is proposed as a significant influencer of bladder behavior and health [1]. This lack of understanding regarding reasons for avoiding public restrooms limits the development of effective interventions and policies to promote and improve bladder health.

The goal of this study was to develop a better understanding of women’s perceptions of individual- and environment-level factors that influence the use of the restrooms at work, at school, or in a public setting. As part of a previous electronic, survey-based study that investigated the associations between lower urinary tract symptoms (LUTS), toileting behavior, and access to public restrooms [2,3,5], this planned, secondary analysis examined responses to open-ended questions about limitations to restroom use using a mixed-methods approach. Our objectives were to identify predominant constructs that may underlie a women’s attitude towards using restrooms and to develop a conceptual model incorporating these themes that can inform future research or interventions that can improve bladder health.

## 2. Materials and Methods

### 2.1. Eligibility and Recruitment

This was a secondary, planned analysis from an International Review Board approved (#171455), cross-sectional, survey-based study of research participants identified through two recruitment referral databases available at our institution [6,7,8]. Between October and December 2017, approximately 106,000 potential subjects from Research Match and the Research Notifications Distribution received a single email advertisement that incentivized women to complete an anonymous, English-only, electronic survey with the chance to win two randomly drawn Apple iPads. Of those, 7594 at least started the survey, for a response rate of 7.4%. No data were available for nonresponders.

### 2.2. Data Collection

As part of the electronic questionnaire, participants answered questions related to demographic and clinical factors, including lower urinary tract symptoms and bowel function. The survey collected self-reported information on age; race/ethnicity (non-Hispanic White, non-Hispanic Black, Asian, other/multinational, and Hispanic); highest education (less than college versus college, graduate, or higher); work status (unemployed or other, part-time, and full-time employed); relationship status (single, married, or divorced/separated/widowed); and parity (0, 1, 2, 3, or more). We also categorized participants by age group (18–24, 25–44, 45–64, and ≥65 years) [1]. Participants reported frequency of bowel movements (categorized as >2 times a day; twice a day, once a day, or every other day; or every 3 or more days). Lower urinary tract symptoms were assessed with the International Consultation on Incontinence Questionnaire Female Lower Urinary Tract Symptoms (ICIQ-FLUTS) with a 4-week recall [9]. Bladder condition bother was assessed with the Patient Perception of Bladder Condition, which is a 1-item instrument that measures global patient-reported assessment of bladder condition based on a scale of 1 to 6 [10]. Women who responded “My bladder condition does not cause me any problems at all” (score 1) were designated as being not bothered, while those with a score greater than 1 (at least “some very minor problems”) were categorized as bothered [3].

Participants were offered open-ended responses to questions about perceived limitations to using the restroom at work and public restrooms. Embedded in each survey were seven free-text fields, which asked people to explain or elaborate on their fixed-choice responses. Whether women limited restroom use at work was assessed by asking “Do you purposefully limit your use of the restroom at work?” (not at all, occasionally, sometimes, most of the time, and all of the time). Participants indicating that they limit restroom use at least occasionally were also asked, “Please select reasons you limit use of the bathroom at work. Check all that apply” with response options of “Quality of the restroom is poor”, “Limited availability of restroom”, “Too busy with work”, “Restricted access by supervisor/company”, and “Other reason”. For each selection, they were offered a free text area to “further explain why…”. They were similarly asked, “Do you purposefully limit your use of the restroom out in public?” (not at all, occasionally, sometimes, most of the time, and all of the time), followed by “Please select reasons you limit use of the bathroom out in public. Check all that apply.” (quality of the restroom is poor; limited availability of restroom; long line to use; and other reason). Again, for each selection, they were offered a free text area to “further explain why…”

### 2.3. Methods for Qualitative Analyses

Qualitative data coding and analysis was managed by the Vanderbilt University Qualitative Research Core, led by a PhD-level psychologist. Data coding and analysis was conducted by following the consolidated criteria for reporting qualitative research (COREQ) guidelines [11], an evidence-based qualitative methodology. A hierarchical coding system was developed and refined using the open-ended survey questions and a preliminary review of the quotations. Major categories included (1) restroom type; (2) quality; (3) psychological factors; (4) perceived accessibility; (5) restrictions; (6) privacy factors; (7) biological concerns; (8) partiality; (9) alternative action; and (10) urgency. Major categories were further divided from one to six subcategories, with some of the subcategories having additional levels of hierarchical divisions. Definitions and rules were written for each category. The coding system is presented in Appendix A.

Experienced qualitative coders independently coded one-fourth of all quotes. Coding was then compared, and any discrepancies were resolved. After establishing reliability in using the coding system, the coders divided and independently coded the remaining uncoded quotes. There were no quotes that were excluded from coding and analysis. Each statement was treated as a separate quote and could be assigned up to five different codes. Coded transcripts were combined and sorted by code. Transcripts, quotations, and codes were managed using Microsoft Excel 2016 and SPSS version 26.0. Analysis consisted of interpreting the coded quotes and of identifying higher-order themes using an iterative inductive-deductive approach [12]. The goal of the iterative inductive-deductive approach was to develop a conceptual framework that is theoretically informed while integrating resultant content from the qualitative data. Deductively, the analysis was guided by Social Cognitive Theory. Inductively, the codes and themes from the quotes were used to fill in the details of the conceptual framework [13,14].

### 2.4. Statistical Methods

Demographic and clinical variables were compared using nonparametric tests (Wilcoxon Rank-Sum) and Chi Square analyses for continuous and categorical variables, respectively. Significance was set at *p* < 0.05, and no corrections were made for multiple comparisons. All statistical analyses were performed using Stata 16 (StataCorp, College Station, TX, USA).

## 3. Results

### 3.1. Cohort/data Description

There were 5397 participants who entered free-text responses that contributed to the qualitative analysis. Demographic and clinical characteristics of this group are present in Table 1. Overall, there were few actual differences between women who did and did not contribute free-text responses. Those who did contribute responses were slightly more bothered by their bladder condition (54% vs. 51%). Respondents also significantly more often limited restroom use at least occasionally at work and in public. Fifty-nine percent of the participants provided only one free-text response, 31% provided two responses, and 10% provided three or more responses. There was a total of 12,583 quotes that were coded and used to construct the conceptual framework.

### 3.2. Conceptual Model Summary

As a result of our qualitative analysis, we developed a biopsychosocial conceptual framework that illustrates that, as a woman senses urgency to empty her bladder or bowels, there are various factors that influence what she does (Figure 1). On the left, personal context, comprised of biological concerns and personal history of restroom experiences, sets the stage for decisions concerning restroom usage in a public facility. On the right, when faced with a restroom situation, women consider situational influences, including cognitions, emotions, challenges, and places, to decide to search for, utilize, avoid, or delay using a restroom. In the center, underlying these decisions is urgency (bowel or bladder)—particularly contingent on the severity of urgency—and the various behavioral strategies that women adopt to manage or limit urgency and to navigate the situational influences that affect her ability and willingness to use either a public, work, or school bathroom. The circular shape represents that the strategies are interchangeable, overlapping, and interactive.

#### 3.2.1. Personal Context

There are concerns centering on bathroom use that are salient for women, are constantly present, and therefore set the stage for how women experience and utilize bathrooms. We identified two major categories that form this personal context: (1) biological concerns and (2) salient personal history associated with public restrooms (Box 1). In terms of biological concerns, there are several health conditions that affect how women experience public restrooms, including digestive, bladder, and other medical conditions. Digestive conditions that result in bowel movements accompanied with gas, noise, or diarrhea are concerns that influence restroom decision making. Bladder conditions can influence a woman’s decision and ability to delay bathroom use by holding her urine, create difficulties urinating, or modify the frequency and urgency of needing to urinate. Other medical conditions, especially those that limit mobility or increase vulnerability to infection, also function as contextual factors that must be considered across specific bathroom situations. Regarding personal history, participants reported that frightening encounters in the past influence their bathroom decisions. These experiences include confrontations with aggressive or angry people, violations of privacy, and hostility related to gender nonconformance.

Box 1Peronal context. Irritable bowel syndrome (IBS), Postural orthostatic tachycardia syndrome (POTS).
***Biological concerns***

*“I have IBS and do not like to defecate in the public restroom at work.”*

*“I do not like to use the bathroom with strangers present, especially when I am defecating. I suffer from ulcerative colitis, and my bowel movements are often embarrassing.”*

*“I have Crohn’s Disease and therefore really noisy explosive diarrhea.”*

*“Because I have been diagnosed with interstitial cystitis, I hate having to wait for the bathroom. There is only one staff bathroom available, and sometimes, it is occupied when I need it.” “I have to catheterize to urinate, and public restrooms make it difficult to set up a sterile environment—from touching stall handles, etc.”*

*“Public restrooms are probably filled with germs. I have immune deficiency, POTS, etc., so I try to limit exposures.”*

*“I have one kidney, and I worry about infections of bladder and kidney.”*

*“I receive an antineoplastic agent [for cancer]… and am fearful of infections”*

***Personal history***

*“I do not feel safe in public restrooms related to a previous scary encounter.”*

*“I have also had adverse encounters in public bathrooms, when individuals made hostile or aggressive comments, which are a deterrent.”*

*“Even though I mind my business, as we all should, I have had so many instances where I have felt very uncomfortable due to others NOT minding their business. So, this is another reason that causes me to limit my use.”*

*“Last time I used a public restroom, there was a man in there peeing with the door open and darted out before washing his hands.”*

*“I was almost lured away in a mall bathroom by a janitor when I was very young. My dad intervened.”*

*“I got incredibly scared by a flushing toilet on a train when I was a small girl which left me with a generalized fear of flushing toilets.”*

*“A family member was murdered in a public restroom.”*


#### 3.2.2. Situational Influences

Any time a woman is away from home and experiences a need to urinate or defecate, she is faced with having to make a series of decisions, such as whether to find a restroom or if an available restroom is acceptable. Our participants reported a wide range of situational influences that are taken into consideration when making bathroom decisions. These situational influences involved cognitions, emotions, challenges, and places. Women reported thoughts they have about what they might encounter if they choose to use a specific bathroom. They also reported a range of negative emotions that are triggered by considering using or actually using a public restroom. Women identified aspects of the situational context, where they are and what they are doing at the time, which place constraints or introduce challenges associated with using a public restroom. There were many reports about the physical characteristics of specific bathrooms that must be considered when deciding whether to utilize a specific public facility.

##### Cognitions

Cognition is the appraisal process that takes into account the current situation and the possibility that the restroom could be risky, the potential of human encounters, and handling one’s standard ruminations and phobias. Women expressed a variety of cognitive influences that control their decision to use a public toilet, including risk of assault, human interactions, and phobias (Box 2). Women were especially concerned with risk of assault. While some may have these concerns based on past personal experience, many had thoughts about the dangers of using a public restroom without a history of having been assaulted. The influence of human interactions was reported, primarily in the workplace. By their very nature, public restrooms are places where there is a high chance of encountering other people as you enter, use, or exit the facility. Woman anticipate these interactions, and those anticipations can lead to strategies to delay or avoid using a public restroom or to seek another restroom where the chances of encountering other people are perceived to be lower. Phobias and ruminations are also associated with public and work restrooms. There are unpleasant, scary, or uncomfortable aspects of public restrooms that participants reported they cannot stop thinking about. The outcomes of the phobia or ruminative thinking are emotional, often triggering anxiety.

Box 2Cognitions. Obsessive compulsive disorder (OCD), Acquired immunodeficiency syndrome (AIDS), Sexually transmitted diseases (STDs).
***Assault***

*“Most look like magnets for perverts and mass murderers.”*

*“Often, bathrooms are in areas that are very isolated or bathrooms are empty, creating situations that can leave women vulnerable to assault.”*

*“Fear of possible voyeurism due to restroom gender policies or unknown cameras placed.”*

***Human interactions***

*“I do not want to run into my colleagues or students in the bathroom.”*

*“I feel awkward being in the restroom at the same time as my coworkers.”*

*“I am in a wheelchair and feel uncomfortable having people watch me go in and out of restrooms. It is not awkward when I am by myself; it is just awkward to be observed because people always feel the need to help me, and I really do not want that when I am trying to do something private like go to the bathroom.”*

*“I usually use the handicap stalls due to setting up catheterizing supplies but I have had people with canes stand outside the door tapping their canes—when I emerge, since I do not look handicapped, I am usually greeted with scowls, just an uncomfortable experience.”*

***Phobia and Rumination***

*“I have a phobia that makes it extremely uncomfortable to be in a public restroom without suffering an anxiety attack.”*

*“I have a phobia of peeing in public if there is someone else in the bathroom.”*

*“I also have mild OCD and in the past have worried about contracting AIDS or STDs from public restrooms.”*

*“I am claustrophobic and do not use one person restrooms where I have to lock myself in for fear the lock will jam and I will get stuck in the restroom.”*


##### Emotions

Emotional responses are generated by cognitions and influence a woman’s decision to use or avoid a public restroom. Participants described a range of negative emotional reactions when faced with the possibility of using the public facility (Box 3). Many reported experiencing fear, anxiety, embarrassment, and disgust. Thoughts of using a restroom can trigger strong feelings of fear and anxiety. The fears are associated with several aspects of restroom use including other people being aware that she is using the restroom and having others listen to her urinating or defecating. Embarrassment, which involves being acutely self-conscious with feelings of shame, was reported many times. Embarrassment comes into play when there are other people near or in the restroom when it is being used. Past experiences of feeling embarrassed can lead to the anticipation of feeling embarrassment. Disgust was a prominently mentioned emotional reaction to public restrooms. Women reported disgust reactions to urine, feces, and stray menstrual blood on the floor and in and around the toilet. Physical interaction with bathroom-related items were also primary sources of disgust responses.

Box 3Emotions.
***Fear and anxiety***

*“There is only one path to the restroom from my desk, and I have to walk past everyone, which I get slight anxiety from if I constantly have to pee.”*

*“When I am at work or using public restrooms, I feel like the urge for bowel elimination does not come to me naturally. I am unsure if my fear or anxiety of other colleagues knowing about my bathroom activity plays a part in this sort of bodily suppression.”*

*“It is really close to desks, and I am afraid people will hear me or judge how long I am taking.”*

*“I have issues peeing in public as I panic when others can hear me urinate. Even if I strain, I cannot urinate once I start to panic.”*

*“No privacy: I am afraid others may hear me doing my business and be offended.”*

***Embarrassment***

*“If there is another person in the restroom or if someone is waiting for me, I cannot pee. It is embarrassing.”*

*“Only one toilet for a lot of people: I am sometimes embarrassed to use it—people usually know who is in the bathroom.”*

*“I have extremely slow, lengthy bathroom visits that can be embarrassing. I usually pass feces even when I only feel the need to urinate, and it can take 20 min or longer, so it feels awkward to do this in a public restroom.”*

***Disgust***

*“Gross: Women peeing on toilet seats is biological warfare.”*

*“People pee on the seats all the damn time, or there will be actual feces on the seat or behind the seat of the toilet and visible on the rim.”*

*“I’m not comfortable going out into public cause women can be so nasty in the bathroom. I saw a tampon and a piece of macaroni in the same toilet one time, and it scarred me for life. Did it come from the same place?”*

*“I do not like the idea of touching something that has been touched by other bottoms. It feels gross. Plus, splash back from the bowl feels like I could be getting their urine on mine. Plus, the space in the front between the two sides of the toilet seat is usually disgusting, and sometimes, my underwear touches it.”*

*“Since I live in an urban setting, many restaurants use keys, which is disgusting. I do not want to handle it, then walk back, and eat food”*


##### Challenges

Beyond thoughts and feelings about restroom use, the participants identified specific environmental features that can make restroom use challenging (Box 4). Time constraints and rules and obligations were significant barriers to bathroom use. These often interact in that rules around work or school are such that they create time constraints, forcing restroom use to occur during a short time window of availability. Women reported deterrence from employers and supervisors to use the restroom outside of breaks. Designated breaks for restroom use are enforced, and at times, women are criticized for using the restroom outside of breaks.

Public restrooms can be crowded. In addition to creating apprehension about using a restroom, crowding increases the length of time required for a bathroom break and can enhance fears and apprehensions about using a public restroom. The number of facilities for the size of the crowd can be too small. At public events like concerts or sporting events, there are almost always lines for the women’s restrooms. Small or poorly designed restrooms can feel crowded and, while not leading to delays, can result in anxiety or other discomforts. When restrooms are crowded, they tend to be less clean, leading to actual or anticipated feelings of anxiety or disgust.

An available restroom or an acceptable restroom a long distance away was discussed as another challenge around restroom use. The distance can cause problems because it increases the time required to use a restroom or can be an obstacle for people with mobility constraints.

Using a restroom with a child creates a number of challenges. Mothers identified finding a bathroom to use when accompanied by a child as a challenge. This becomes a more difficult problem when an infant needs to have a diaper change. It can also be a problem when accompanied by a young child of the opposite sex who needs assistance or supervision.

Box 4Challenges.
***Rules and Obligations***

***School settings:***

*“At my school, one of the buildings has male bathrooms on the first floor and female bathrooms on the second floor. If a female wants to use the restroom during class, she has to miss a large chunk of time to even get to the bathroom in the first place.”*

*“Often in a classroom setting where leaving class would mean missing lecture and compromising my grade”*

*“During class, it is discouraged to leave the classroom.”*

***Work settings***

*“I can usually only go during my breaks and must ask my supervisor for permission if not during a break, which is embarrassing.”*

*“We are told to keep our bathroom use to an absolute minimum in order to be more productive: not a written rule, of course, just a verbal reminder to everyone.”*

*“I have been reprimanded for using the bathroom too often.”*

*“My bathroom breaks are considered ‘personal time’ and I have to choose that disposition on my computer (I work in a call center). This is calculated into my monthly score and compared to other coworkers. If we take too much ‘personal time’, we will be asked to limit it. If I really hurry, it takes me 3.5 min to walk from my desk to the bathroom, pee, wash hands, walk back, unlock my computer, and choose the ‘available’ disposition. I hold it until my designated break but that varies as it is not a set time but we go in order. We are also supposed to communicate with our coworkers via messenger when we are taking personal time and when we have returned from personal time. I do not like telling 16 other adults when I am going to pee.”*

*“If I use the restroom too much, I could get in trouble/fired because they will think I am not doing my job.”*

***Crowding***

*“The stalls are small, and I have to put my bags on the dirty ground. During winter, if there is no hook in the stall, I will be peeing with my hat, scarf, and coat on”*

*“The room is small, cramped, and poorly laid out.”*

***Distance***

*“Most stores have restrooms in the back of the store, I have trouble walking long distances, and if I need to go to the bathroom, I sometimes do not make it there without a leak.”*

*“I cannot use bathroom on my floor.”*

*“It is a far walk from my desk; it takes 6–10 min.”*

*“The area in which our office is located does not have a private bathroom. One must leave the location and go into another building to use the bathroom.”*

***Parenting***

*“I have a 14 month old, and I often carry her into stores. If I have to use the restroom, I have nowhere to put her while I go.”*

*“I have a baby, and every time I need to use the large stall with the changing table, some person is in there who does not have a baby and is not disabled!”*

*“There is no family restroom and my 7-year-old son does not want to go in the women’s restroom, and I do not feel comfortable letting him use the men’s room alone.”*


##### Places

Not all public restrooms are the same. The specific characteristics of a restroom affect a woman’s willingness to use that restroom or to avoid or delay restroom use (Box 5). There was much discussion around the cleanliness of restrooms, and many specific complaints about how bathrooms are unclean or unsanitary. Uncleanliness is characterized by visible feces or urine on the floor, walls, or toilets. The smell can also be a signal that a restroom is unclean. Women reported that unisex facilities are a problem because men will urinate on the toilet seat or dribble on the floor. Restrooms that are poorly maintained, untended, littered with trash, or infested with insects are viewed as unclean.

Bathrooms vary in their amenities such as locking stall doors, soap, hand sanitizer, well-supplied paper products, ventilation, and availability of seat covers. Lack of amenities can be compensated for by bringing one’s own supplies. However, inadequate, substandard, missing, or poorly maintained amenities can lead to restroom avoidance.

Smells and sounds are problems in two ways. First, women do not like to hear others using a bathroom or to be exposed to other people’s odors. Many are also uncomfortable using a restroom when another woman can hear or smell what they are doing. Odor can also be a more general problem when a bathroom is not well maintained. Different restroom layouts and ventilation influence the intensity of odors and sounds when using them.

Bathrooms are often seen as dangerous or unsafe. Women’s concerns about safety was primarily the possibility of assault. Location of a bathroom, the surrounding area, and how many people are nearby influence perceptions of how safe it is to use a specific facility. There were also concerns that men could by spying on them or secretly recording their bathroom use. Safety concerns vary by location and are mainly associated with the use of public restrooms. Poor lighting, remote location, few people nearby, and time of day also influence perceptions of safety.

The sense of privacy when using a facility is very important, with a more private setting preferred. Privacy is a very important aspect of a restroom that is considered when deciding whether to use a specific facility. Privacy can mean using a restroom alone so that others cannot see, hear, or smell what you are doing. Some women only truly feel comfortable and private at home and do not like to use any public restrooms.

For people with mobility limitations, the presence of accessible facilities is very important. Problems with accessibility were reported for public and work restrooms. Many restrooms can be difficult to navigate for people with special needs. Another accessibility concern is the lack of sufficient facilities for women compared to men.

Box 5Places.
***Cleanliness/sanitation***

*“In a unisex restroom, men had peed all over the seat, floor, and back of the toilet.... I thought he had to have been blind or drunk.”*

*“Smells awful and sanitary waste disposal containers were not emptied for the last 6 months”*

*“Since it has not been cleaned well (urine on the seat and floor which then dry and leave stains), I feel like I am dirty when I leave the bathroom.”*

*“Our restrooms are filthy with dead roaches, and there is no ventilation system present.”*

***Amenities***

*“Sometimes, they have run out of hand soap. Sometimes, it is just not very clean. There are no covers on the toilet paper and no lids on the toilets so the spray can go all over. Sometimes, they run out of toilet paper. Sometimes, the toilet is leaking.”*

*“The water takes forever to get even tepid. It smells of sewage. The fan-thing in ceiling is very loud, and the faucet is broken.”*

*“The auto flush is overactive and splashes water all over my bottom when I am urinating.”*

*“The paper towel dispenser is broken so we just have a big roll sitting around that we have to pull from. The utility sink faucet leaks regularly. Hand washing sinks back up periodically (we are in old building with very old plumbing!).”*

***Odor/sounds***

*“The smell can bother me. If the restroom smells strongly of urine, feces, or bleach, I will either become nauseated or have difficulty breathing.”*

*“Sometimes, the bathrooms smell bad and it becomes a task trying to squat AND hold my breath at the same time.”*

*“At times, the women’s bathrooms have a smell of used tampons or pads.”*

*“I have a strong gag reaction to unclean bathroom smells and especially when bleach-y or floral cleaning product smells are just layered on top of them”*

***Privacy***

*“I am uncomfortable with other people, regardless of what my relationship and general level of comfort around them may be, (whether they are friends, family, coworkers, peers, acquaintances, or random strangers), being so closely and directly involved with and exposed to what I consider to be my very private, intimate, personal, and discreet ‘matters of business’, and vice-versa, even if they are perfectly comfortable with the situation (which surprisingly and unfortunately is often the case).”*

*“The walls are *so* thin. I can listen to and understand people talking in the hallway while I am using the bathroom. It is uncomfortable to think what the conversationalists can listen to from the hallway.”*

*“Need privacy for smelly and loud shits”*

***Accessibility***

*“I use a scooter and have limited mobility out of it. All too often, public restrooms’ ‘accessible’ stalls are not big enough that I can turn my scooter around in them.”*

*“I am disabled on the right side, both arm and leg, and have problems at times due to the handicap bar being on the right side of the stall, which is of no help to me at all.”*

*“I have a service dog, and public bathrooms often do not have room for him.”*

*“There are only 2 stalls for women on my floor and many more women than men that work in my area.”*

*“There are more than 100 people who work in our department; we share a bathroom. Most of us are women. There are frequently lines.”*

*“Very distant due to men’s restrooms only being available nearby”*

*“Cannot go if there is not one…:) I am willing to use the men’s room if needed.”*

*“There is ONE (single-person) female restroom and ONE male for both the pharmacy/store/clinic employees and customers. Often, if the women’s [restroom] is occupied, I will just use the men’s restroom because it really does not matter and I do not have TIME to wait for it to become available.”*


#### 3.2.3. Strategies and Solutions

The participants described a number of strategies that women use to manage the multitude of situational influences that affect her ability and willingness to use either a public, work, or school toilet. The strategies are (1) resist and delay using a bathroom; (2) plan ahead to reduce the need for a bathroom or to know where an acceptable public restroom can be found; (3) manage sounds when using a bathroom; (4) restrict fluids in order to avoid the need to use a bathroom; (5) seek alternative locations where a more acceptable bathroom can be found; and (6) bring supplies like wipes and hand sanitizer to insure a safer and more comfortable bathroom experience. These strategies are used to manage or make the experience more acceptable, to reduce risks, or to reduce anxieties. The majority of these strategies are modified by the level of urgency a woman is experiencing. As urgency increases, strategies must be adjusted and eventually may lead to using a bathroom that would otherwise be unacceptable.

##### Resist and Delay

Women described using the strategy to resist and delay relieving herself when faced with a bathroom situation that made them uncomfortable (Box 6). How long a woman can delay and whether she will give in and use a public restroom depends upon both her level of urgency and the conditions of the restroom.

Box 6Resist and Delay.
*“It is as if people have no idea how to function in a public restroom. It is truly disgusting what one comes across. Seats are covered in piss, toilets have not been flushed, paper towel is all over the grounds, and pads have not been wrapped. It is disgusting. Far more often than not, I prefer to hold it.”*

*“There is only 1 women’s restroom (single-person) where I work. I am the only female out of 30 employees where I work, so the men use it frequently, especially for going #2. I refrain from using it because the bathroom is not near my desk, and it is usually occupied or smells bad.”*

*“If restroom is unclean, I may delay urinating if the urgency is low. If the need is higher and it would be distressing not to urinate, I will use the restroom even if it is not clean even though it makes me uncomfortable.”*

*“If and only if I do not need to pee immediately AND I know that I will be able to use my home bathroom, then I balance the need to pee versus the risks associated with germ exposure. Most times, however, the need to pee wins.”*


##### Planning

Planning ahead allows women to avoid unpleasant bathrooms by knowing where acceptable bathrooms are located or by relieving herself before she leaves her home (Box 7). Planning ahead includes being familiar with the location of acceptable public restrooms. Managing the timing on one’s need to eliminate is also a way of planning. This includes strategies such as using the restroom before leaving home, using an acceptable restroom when urgency is low, or timing one’s day to be near a restroom when it is likely to be needed. Sometimes, plans can be elaborate, such as carrying a portable toilet while traveling.

Box 7Planning.
*“When you are in public, you never know when or where there will be a restroom available. That causes a lot of uncertainty and makes you have to plan around whatever you are doing.”*

*“Porta-Potties are usually awful to use, and I will plan around it to avoid using if I can.”*

*“The urgency I have with using the toilet and the comfort at using a commercial bathroom directly influences where I go, where I shop, the time of day. and the length of time spent.”*

*“I time and plan my trips away from home to never be away from a toilet for more than an hour.”*

*“Usually at crowded events, I avoid using the restroom because it is very crowded, so I try to go before the event starts and afterward unless it is an emergency.”*

*“I am more aware and always locate a toilet around work/school/volunteer, etc. where I can go in an emergency. I know the best restrooms in 3 cities around the USA.”*

*“I am in a wheelchair and need an attendant. It can make the average restroom a luxury. I have started carrying a portable toilet as we travel a lot.”*


##### Managing Sounds

Women had several ways of managing sounds of urinating and passing gas in order to reduce or avoid unpleasant bathroom experiences (Box 8). Ways of managing sound include creating noise, such as turning on the water or blocking noise. Some will select restrooms with masking noise, such as recorded music playing in the background.

Box 8Managing Sounds.
*“I sometimes turn on the water to disguise the noise of the flow if I think someone could hear outside of the bathroom (would not apply if using a stall).”*

*“If there are other people in the public restroom, I usually cannot go no matter how hard I try. I have a shy bladder. Sometimes, I can get over that by putting my fingers in each ear and wiggling them around so I cannot hear myself pee.”*

*“I do not like to hear others pass gas while they urinate, it embarrassed me to hear it, so I try to avoid being in room with others.”*

*“It is embarrassing that I am embarrassed to make the sound and embarrassing that I am embarrassed to NOT make the sound. I like when restrooms have fairly loud music playing in them to mitigate this dumb fear/habit of mine.”*


##### Restrict Fluid

Women also reported restricting fluids at work and when away from home in order to avoid or to delay the need to urinate (Box 9). Many restrict fluids in order to prevent the disruption of workflow, frequency of urination in social settings, use of undesirable toilet environments, and delaying the need to the search for a bathroom.

Box 9Restricting Fluids.
*“I have to urinate A LOT whenever I am normally hydrated, so I purposefully do not hydrate during the day so I am not interrupting my work by getting up to use the restroom every 30–60 min.”*

*“If I go to a restaurant, I may have to pee like 5 times during the meal and that can be embarrassing, so I will stop drinking (drinking as in beverage, not drinking as in alcohol) as much so I do not have to keep peeing.”*

*“If they are not around or my only option is a portable bathroom, I will try to hold my urine as long as possible and limit my intake of fluids.”*


##### Alternate Location

Seeking alternative locations was used for numerous bathroom situations (Box 10). The primary explanations for seeking alternative locations are due to privacy concerns, unsanitary conditions, and long lines.

Box 10Alternate Locations.
*“I travel a lot for work, and I do not like using public bathrooms because of cleanliness and privacy concerns. I try to stop by my house or a friend or family’s if at all possible.”*

*“If the bathroom is too dirty (i.e., urine or feces on the seat, or no one has flushed the toilet), I will hold it until I am in a cleaner bathroom.”*

*“If the bathroom is extremely dirty, I may drive down the street to another establishment.”*

*“I know I CANNOT wait for longer than a few seconds; I will choose to pee in my pants in privacy as I walk to the car. It may result in me having to sit down on steps to minimize visual cues to onlookers…. A person experiencing a bladder with its own mind has to gauge the risks and challenges within seconds… a cough which causes urine detection IS embarrassing and unprofessional. A bladder incident which creates unpredictable showers soaking the pavement and clothes is another. Welcome to our world. One day, just a wet spot... the next, an embarrassing shower on the pavement.”*

*“If I know I am going to be going somewhere else soon with a nicer restroom, I will wait as long as it does not get to a point where I feel extreme urgency to go. If I really need to go, I will go anywhere.”*

*“I still NEED to use it, but it is difficult to wait for an open stall. Historically, I and other women will ‘keep watch’ while we take turns using the men’s restroom. (Sometimes, in using the men’s restroom, ‘filthy’ is at an all-time high.)”*


##### Supplies

Some participants reported bringing their own supplies to the restroom in order to manage the potential and the reality of germs (Box 11). Women reported bringing gloves, wipes, paper, hand sanitizer, or water bottles to use as bidets.

Box 11Supplies.
*“Often, the restroom is not clean. It has a foul odor and no toilet seat covers. I often bring cleaning wipes and gloves to clean the seat before I lay tissue on the seat so I can sit down (bad knees) as well as use the handicap stall to hold on to the bar while I squat.”*

*“I find it very stressful when toilet water does get on me. I try to carry antibacterial wipes or hand sanitizer with me in case this happens. I wish that hand sanitizer was available inside the stalls.”*


## 4. Discussion

Toileting behaviors are often considered to be a private matter, and therefore, barriers to restroom use at work and in public are frequently not discussed and have been minimally studied. This qualitative study evaluating open-ended responses of perceived limitations to restroom use at work and in public reveals a complicated interplay of personal contexts, situational influences, and behavioral strategies used by women to manage their bladder and bowel habits away from home. This interplay supports emerging conceptual models of how social ecological and environment influences strongly determine behaviors and habits that embody biological changes impacting on a woman’s bladder health across her life [1]. For the most part, such factors have been ignored in the study of LUTS, which has tended to focus historically on characterizations of symptoms and, to a lesser extent, quality of life primarily in women with recognized lower urinary tract dysfunction (e.g., urinary incontinence or overactive bladder syndrome) and not on women without urinary dysfunction (i.e., “healthy” women). A search of the medical literature really reveals a dearth of studies that have examined the relationship between bladder health and restroom use outside the home.

To our knowledge, this is the largest study to date using open-ended responses to develop a conceptual model of the limitations to public and workplace restroom use. Other recently published studies have used focus groups of women to investigate the interrelationship between conscious decision-making processes and bladder sensation in determining the need, time, and place to void [13] and women’s experiences navigating toilet access and use [15,16]. Harvey et al., using interview focus groups to study a total of 25 women, developed a thematic analysis with six themes (temporal and cognitive maps, risk issues, habituation and opportunistic behavior, and awareness of the bladder) surrounding a woman’s determination of the need, time, and place to void [13]. The authors found that, most of the time, the decision to void was determined by integrating multiple factors aside from the simple sensation of bladder filling. There are similarities between the themes noted by Harvey et al. and the behavioral strategy constructs identified in our study including planning ahead, alternative locations, and managing sounds. Similar to our study, Harvey et al. also identified cleanliness and safety concerns as contributing factors to the cognitive decision of when and where to use the restroom.

The Prevention of Lower Urinary Tract Symptoms (PLUS) Research Consortium conducted 44 focus groups with a total of 360 participants to identify factors influencing perceptions, beliefs, and behaviors around women’s toilet access [15]. The investigators identified several themes that were similar to those in the present study, including the concepts of “gatekeepers” (i.e., individuals who control access to toilets) as a barrier to restroom use and self-restricting toilet use (i.e., deciding not to use the toilet despite biologic need to urinate) in an effort to maximize learning time at school or productivity at work or to avoid unclean, odorous, or dirty restrooms. In our contextual framework, gatekeepers are included in the challenges under situational influences, while self-restricting behavior is categorized as a behavioral strategy.

Palmer et al. recently performed a focus group study of 24 women to better understand women’s knowledge about bladder health and urination needs [16]. The authors identified common themes influencing urination needs among participants, including cues/triggers/alerts (both internal and external), cleanliness of facilities, the nuisance of toileting, and situational awareness. Overall, these themes are also represented in our contextual framework; however, our framework provides a more organized way of classifying personal and situational influences and behavioral strategies surrounding a woman’s toileting behaviors.

The common themes within personal contexts identified in our study reveal biological concerns and personal experiences that are unable to be modified by public restroom design or workplace environment and practices. In other words, the personal context is intangible and is something unique to each individual, and regardless of improvements made to public restrooms and workplace restroom use policy, these personal contexts may still persist and may continue to be a limitation. There are, however, many opportunities for improvement in public restroom design that could help to mitigate the limitation of these personal contexts [17]. Public and workplace restrooms could be placed in easily accessible areas such that a person with digestive conditions or urgency may be able to reach a restroom quickly should the need arise. Additionally, panic buttons could be added to restrooms for emergency situations so that a person with a personal history of a traumatic memory or frightening experience could call for help if needed. These measures may be able to add a sense of security to restroom use.

Similarly, situational influences of cognitions and emotions may also still persist regardless of restroom design and practices but can be limited by implementing new changes. For example, soundproof walls and doors may help to decrease embarrassment surrounding restroom use at work by limiting the noises heard by others close to the restroom [18]. Recurring cognitions of risk of assault and safety concerns can be decreased by well-lit hallways, security cameras, and emergency call buttons.

Our study found time constraints, and rules and regulations to be common limitations to restroom use at work. Prior studies have shown that “waiting too long to urinate” is associated with LUTS in women [19,20] and that women in occupations where restroom use is limited have a higher prevalence of LUTS compared to those with less restricted restroom access [5,19,21]. Increased education for employers and changes to the way time for restroom use affects interpretation of “productivity” by employers is necessary in order for women to be better able to manage bladder health in a work environment.

Previous work has been done in the greater workplace safety realm to develop a sociotechnical model with concentric layers of the work system, socio-organizational context, and the external environment [22]. Our study has similarly revealed the complex interaction of personal and situational influences with behavioral strategies for managing restroom use at work and in public. There is a challenging interplay between ensuring privacy and decreasing scrutiny for those that need to use the restroom frequently at work while also providing a safe toileting environment and decreasing fear associated with isolated restrooms. All of these contexts and influences will need to be taken into consideration when developing functional recommendations for improved restroom design and work policies in order to promote healthy bladder habits.

Our study is the first of its kind to use open-ended responses to a cross-sectional, survey-based study to develop a conceptual framework of limitations to women’s restroom use. Previous studies utilizing focus groups have vastly fewer participants and therefore cautious applicability to the greater population; however, our study has found similar themes to previously published studies. The nature of our qualitative research limits its objective comparison to previously performed studies; however, similarities between our research and other published qualitative studies utilizing focus groups is reassuring that the contextual framework developed in the present study utilizing a large number of open-ended responses holds validity.

There are several limitations of this study. Our overall response rate was only 7.4%, and self-selection of participants, especially those that took time to respond to the open-ended questions, may influence our results. We only solicited participation with a single advertisement; likely with repeated advertisements, we might have been able to improve the response rates. This rate is consistent, however, with previous studies using this and similar resources with electronic advertisement [6,8]. We also did not target specific demographic groups, such as age or race/ethnic groups, which likely resulted in a cohort that was skewed toward younger and non-Hispanic White participants and thus less generalizable to the population at-large. Additionally, the length of the survey may have prevented some respondents from completing the survey, although our pilot testing suggested the burden was still quite low for most women. Further response bias may reflect that responding participants may be a subpopulation of women with more barriers or limitations to restroom use and thus more likely to supply free-text responses. Due to the nature of this study’s methods, follow-up questions could not be asked of the participants and hypothetical alternative environments could not be discussed with the participants to see how they may influence their toileting behavior choices. We also did not separate women’s responses based on whether they had LUTS or bladder bother, which may identify additional themes that could inform our conceptual model. Future studies are needed to confirm our findings and to expand on the themes identified. However, regardless of these limitations, this study still reveals the personal and situational influences and behavioral strategies used by many women to decide when and where to use the restroom. Some women may not encounter any of these limitations, and it is important to remember that each individual can have personalized influences within the framework that affect her restroom use habits.

## 5. Conclusions

A woman’s decision to search for, utilize, or avoid restroom use at work or in public to empty her bladder/bowels is based on a complicated interplay of personal contexts, situational influences, and personal behavioral strategies. These findings are significant because they reinforce the notion that factors beyond immediate biological needs often influence restroom use—factors that are rarely considered in most intervention studies, for which simple frequency of daily voids or incontinence episodes are the primary endpoints. Future studies are needed to better understand whether these personal contexts or behavioral strategies can be directly targeted as interventions or as patient-centered outcome measures or both. Importantly, these findings can also inform future research and public policy with regard to toilet access in the workplace and in public and how it relates to bladder health awareness.

## Figures and Tables

**Figure 1 ijerph-17-02053-f001:**
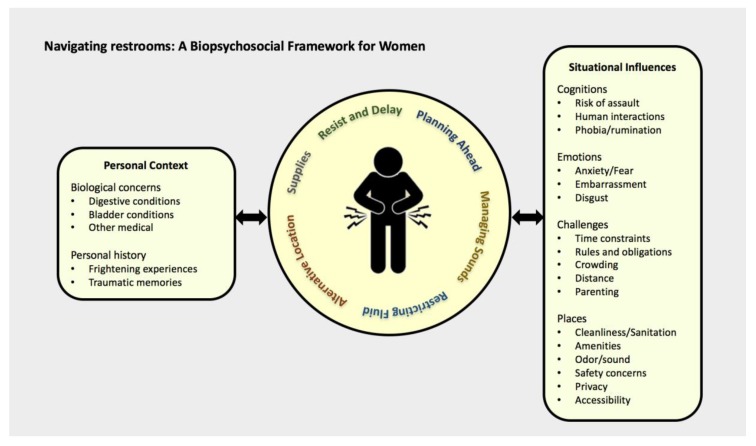
A biopsychosocial framework for women navigating restrooms.

**Table 1 ijerph-17-02053-t001:** Demographic and clinical information of women entering free-text responses that informed the qualitative analysis, compared to those who did not enter any responses.

	Provided at Least One Free-Text Response		*p* Values
	No	Yes	
Number (%)	2197 (29)	5397 (71)	
Age, median (Interquartile range)	38 (28–55)	37 (28–53)	0.23
Age Group (years), *N* (%)			
18–24	318 (14)	663 (12)	<0.001
25–44	1006 (46)	2700 (50)	
45–64	647 (29)	1643 (30)	
≥65	226 (10)	391 (7)	
Race/Ethnicity, N (%)			
Non-Hispanic White	1781 (82)	4201 (78)	<0.001
Non-Hispanic Black	131 (6)	473 (9)	
Asian	96 (4)	210 (4)	
Other/Multiracial	81 (4)	248 (5)	
Hispanic	91 (4)	259 (5)	
Education, N (%)			
Less than college degree	685 (31)	1609 (30)	0.24
College degree or higher	1512 (69)	3788 (70)	
Employment Status, N (%)			
Unemployed or other	658 (30)	1569 (29)	0.53
Part-time	396 (18)	944 (17)	
Full-time	1143 (52)	2884 (53)	
Relationship Status, N (%)			
Single	898 (41)	2145 (40)	0.186
Married	942 (43)	2433 (45)	
Divorced/separated/widowed	357 (16)	819 (15)	
Parity, N (%)			
0	748 (48)	2451 (47)	0.17
1	190 (12)	729 (14)	
2	291 (19)	885 (17)	
3 or more	334 (21)	1129 (22)	
ICIQ FLUTS, median (Interquartile range)	6 (3–10)	7 (4–10)	0.005
Bothered by bladder condition, N (%)	1015 (51)	2890 (54)	0.02
Bowel Movement Frequency, N (%)			
>2 times a day	204 (13)	544 (10)	<0.001
Normal (Every other–2 times a day)	1264 (80)	4166 (79)	
Every 3 or more days	122 (8)	556 (11)	
Limit Restroom use at work, at least occasionally *	232 (19)	2626 (68)	<0.001
Limit Public Restroom use, at least occasionally	285 (17)	5311 (98)	<0.001

* This was only asked of women who reported they worked or volunteered outside the home or were students (total *n* = 5066). International Consultation on Incontinence Modular Questionnaire on Female Lower Urinary Tract Symptoms (ICIQ FLUTS).

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
