# Peer review of "Why Do Women Not Use the Bathroom? Women’s Attitudes and Beliefs on Using Public Restrooms"

_ijerph, 2020, doi:10.3390/ijerph17062053_

Round 1

Reviewer 1 Report

This was a secondary analysis of qualitative data collected from a cohort of women enrolled in an online survey study assessing women’s perceptions of individual and environmental factors influencing public restrooms.  Open ended questions were collected to develop themes surrounding attitudes toward using restrooms at work, school and other public places.

Overall the authors should be commended on the qualitative approach and use of the mixed methodology to better understand the complex interactions surrounding toileting behaviors in women.  This work builds on novel efforts to better understand bladder health and barriers to healthy toileting behaviors.  This work is novel and important; however the data and interpretation are limited by biases that are mostly addressed in the limitations section of the discussion and areas for future research.  Perhaps recommendations in the discussion on how to overcome these limitations will help others design future studies to expand on the field.

a).Online only recruitment strategy severely limits the generalizability of the findings to millions of women without access to computers or interest in online research. Additional studies using alternative recruitment strategies, populations and modes of survey administration should be conducted to confirm these findings.

b). The extremely low response rate of only 7.4% overall and only 68% of these provided responses from this already highly selective population is problematic.  How would the authors approach improving response rates? 

c). Suspected lack of description of sociodemographic diversity. Additional information regarding the demographics of those included in this analysis would be helpful to better understand the population and generalizability of results (e.g. age, parity, race, gender identity, education, socioeconomic status, etc).  Please provide a summary of what data were collected for this subgroup. 

e). Selection bias related to the fact that women with symptoms or problems are more likely to have responded to the open-ended items than those with no issues (the “yelp” phenomenon). 

The biopsychosocial framework seems to be biased towards those with symptoms.  Many of the comments and experiences provided by the women seem to be LUTS focused.  Were the themes the same for those who denied any bladder problems (e.g. healthy bladders)?  It would be interesting to separate out if those women who did not report any LUTS had similar experiences and cognitions to those with symptoms. 

Given the novel approach, important foundation and potential to build on this work to inform policy and practice changes this deserves publication with minor revisions/responses. 

Author Response

Please see the attachments for:

  • updated manuscript with tracked changes

Below are responses to the reviewer:

Reviewer 1:

This was a secondary analysis of qualitative data collected from a cohort of women enrolled in an online survey study assessing women’s perceptions of individual and environmental factors influencing public restrooms.  Open ended questions were collected to develop themes surrounding attitudes toward using restrooms at work, school and other public places.

Overall the authors should be commended on the qualitative approach and use of the mixed methodology to better understand the complex interactions surrounding toileting behaviors in women.  This work builds on novel efforts to better understand bladder health and barriers to healthy toileting behaviors.  This work is novel and important; however the data and interpretation are limited by biases that are mostly addressed in the limitations section of the discussion and areas for future research.  Perhaps recommendations in the discussion on how to overcome these limitations will help others design future studies to expand on the field.

REPLY:

We have followed the reviewer’s advice and added comments for this to the discussion.

a). Online only recruitment strategy severely limits the generalizability of the findings to millions of women without access to computers or interest in online research. Additional studies using alternative recruitment strategies, populations and modes of survey administration should be conducted to confirm these findings.

REPLY:

We agree. The current method used a convenience sample of women that were recruited with the limitations noted. Our hope would be to confirm these findings in different populations of women.

b). The extremely low response rate of only 7.4% overall and only 68% of these provided responses from this already highly selective population is problematic.  How would the authors approach improving response rates?

REPLY:

Because this was an anonymous, incentivized survey distributed electronically, we only sent a single email soliciting participation. Our rationale was based in part on the concern that individuals may try to complete the survey multiple times to increase their odds of winning the prize. Likely this was not as big a concern as we thought. Therefore we could have improved the response rate by re-distributing the invitation additional times.

c). Suspected lack of description of sociodemographic diversity. Additional information regarding the demographics of those included in this analysis would be helpful to better understand the population and generalizability of results (e.g. age, parity, race, gender identity, education, socioeconomic status, etc).  Please provide a summary of what data were collected for this subgroup.

REPLY:

We agree with the reviewer. We have included a table with demographic information, comparing those who provided responses to those who did not.

e). Selection bias related to the fact that women with symptoms or problems are more likely to have responded to the open-ended items than those with no issues (the “yelp” phenomenon).

REPLY:

We agree with this supposition. From our new data table, there were very few differences between women who did and did not respond to the open-ended items when we compared LUTS and bladder condition bother.

The biopsychosocial framework seems to be biased towards those with symptoms.  Many of the comments and experiences provided by the women seem to be LUTS focused.  Were the themes the same for those who denied any bladder problems (e.g. healthy bladders)?  It would be interesting to separate out if those women who did not report any LUTS had similar experiences and cognitions to those with symptoms.

REPLY:

The reviewer raises an interesting point. For this analysis, we did not separate out those with and without LUTS. This could be examined in future analyses.

Given the novel approach, important foundation and potential to build on this work to inform policy and practice changes this deserves publication with minor revisions/responses.

REPLY:

Thank for the support. We too think this is important work.

Reviewer 2 Report

It is very interesting paper that uses qualitative methodology appropriately. I think that the paper can be published. However, the following revision are required.

First, there is a need for a theoretical part that reviews previous research on use of public restroom.

Second, it is necessary to clearly present what is the method for securing objectivity in the coding process.

Third, each concept is inductively distinguished. A schematic model for causal relationship between these concepts is needed. In basic theory, we need to consider axis coding in grounded theory.

Fourth, the conclusion is very short. It is necessary to add the theoretical significance of this study or possible future studies.

Author Response

Please see the attachment for:

  • updated manuscript with tracked changes

Response to reviewer is below:

Reviewer 2:

It is very interesting paper that uses qualitative methodology appropriately. I think that the paper can be published. However, the following revision are required.

First, there is a need for a theoretical part that reviews previous research on use of public restroom.

REPLY:

There is very limited literature on this topic, from a medical standpoint. The most relevant studies which inform the present manuscript are reviewed in the discussion. We have added additional review material to the introduction and discussion, including reference to emerging concepts that frames bladder health within a social ecological model.

Second, it is necessary to clearly present what is the method for securing objectivity in the coding process.

REPLY:

First, the coding system was reviewed by the entire investigative team and we reached agreement to use it after this review. 

When we started the coding process, we double coded the participant responses (25% of them) and then reconciled any differences.  Because of the volume of data, once the coders had established reliability, the remaining responses were coded by a single coder.

The process of comparing the results of two coders and discussing and reconciling any differences is how we ensure reliability in the coding process.

This is included in the manuscript in lines 103-107. 

Third, each concept is inductively distinguished. A schematic model for causal relationship between these concepts is needed. In basic theory, we need to consider axis coding in grounded theory.

REPLY:

We did not approach the qualitative analysis using a ground theory approach. Instead, we used an iterative inductive/deductive approach.  This is explained in lines 110-116 with citations. This occurs just before the reviewers comments. 

Figure 1 is the conceptual framework that was created using this inductive/deductive approach. This would satisfy the reviewers request for a schematic model.   This framework does show relationships between personal context, situational context and behaviors.  We would not claim that we have identified causal relationships from these qualitative data.  We are explicit in identifying social cognitive theory as our deductive approach. Inductively, we used the coded responses to understand themes and relationships.

Fourth, the conclusion is very short. It is necessary to add the theoretical significance of this study or possible future studies.

REPLY:

We have expanded this section to add theoretical significance of this and future studies, as suggested by the reviewer.  

Round 2

Reviewer 2 Report

Author revised all of thing reviewer commented